# Models with a Cause: Causal Discovery with Language Models on Temporally Ordered Text Data

**Bruce Rushing**                                                        *ejf9db@virginia.edu*
*Research Computing*
*University of Virginia*
*Charlottesville, VA 22901*

**Javier Gomez-Lavin**                                                   *jglavin@purdue.edu*
*Department of Philosophy*
*Purdue University*
*West Lafayette, IN 47907*

**Reviewed on OpenReview:** *https://openreview.net/forum?id=YJddclPGuY&nesting=2&sort=date-desc*

## Abstract

While language models (LMs) have been proposed for causal discovery tasks, it remains unclear whether they possess the inductive biases necessary to identify causal structures in token generation processes. We investigate whether LMs can learn the causal structure governing how tokens depend on their predecessors by testing if they possess the temporal and statistical properties required for causal discovery. We prove that existing algorithms can recover a unique causal model when token sequences satisfy standard causal assumptions and have temporal ordering. LMs' sequential processing and positional encodings enable them to leverage this temporal information. Using controlled experiments on synthetic data generated by mixtures of Markov chains, we test whether LMs learn conditional independencies and Markov exchangeability properties necessary for causal discovery. We find that transformers successfully learn these properties, achieving this not by approximating exact probability distributions but by learning qualitative probability rankings. These synthetic experiments provide initial evidence that LMs possess inductive biases suitable for discovering token-level causal structures.

## 1 Introduction

Recent work has explored using language models (LMs) for causal discovery in code (Li et al., 2024b; Gupta et al., 2023) and natural language (Zhu et al., 2024; Ban et al., 2023). However, a fundamental question remains: do LMs possess the inductive biases necessary to learn causal structures from text data? This question matters because conventional causal discovery algorithms (PC, IC) cannot scale to high-dimensional text data. Moreover, impossibility results (Bareinboim et al., 2022) show that probabilistic models cannot perform causal discovery without appropriate inductive biases. This paper identifies conditions where causal discovery is theoretically possible and tests whether LMs exhibit the required biases. Crucially, we study the causal structure of token generation processes—how words depend on previous words—not the extraction of semantic causal claims from text content (e.g., 'smoking causes cancer'). Understanding token-level causal structure provides a foundation for higher-level causal reasoning tasks.

Consider the sequence of words ["the", "cat", "sat"]. We ask whether the model learns that the word "sat" depends only on the word "cat" given the causal structure, i.e. altering "cat" changes the distribution from which "sat" is drawn. We are not extracting the semantic claim that cats cause sitting. Token-level causation is analogous to how Markov chains describe state transitions rather than semantic relationships between states.

Our approach leverages a key property of text: human language production is often temporally ordered, with words following their predecessors in time. While exceptions exist (e.g., heavily edited documents), temporal ordering is prevalent in typical training corpora. For example, the text of a recorded conversation will have words follow one-another temporally. LMs use positional encodings that enable them to detect token ordering, including temporal sequences. This contrasts with traditional causal discovery methods that operate on unordered tabular data.

We prove that temporal information enables unique causal model recovery under standard assumptions (Markov, minimal, faithful, no hidden confounders). Specifically, temporal ordering resolves the equivalence class produced by conditional independence-based algorithms like PC down to a unique model. Thus, causal discovery from text requires two inductive biases: learning conditional independencies and detecting temporal order. While LMs can encode temporal information through positional encodings, the key empirical question is whether they learn the conditional independence patterns or symmetry properties necessary for causal discovery. Testing on natural text is challenging because the true causal structure is unknown. We therefore use synthetic datasets with known causal structures as a necessary baseline: if LMs cannot learn these patterns in controlled settings, they cannot learn them in real text. We find that Transformer variants can identify the required probabilistic properties under specific conditions, suggesting they possess the inductive biases for causal discovery. Surprisingly, models achieve this without accurately approximating the data distribution—they learn *qualitative probability rankings* rather than exact probabilities.

Our contributions are: (1) We prove temporal information enables unique recovery of certain causal structures, (2) we test four architectures on synthetic Markov chain mixtures for required inductive biases, (3) we demonstrate that Encoder-Decoder, Decoder-only, and Switch transformers learn Markov exchangeability, and (4) we show models learn qualitative probability rankings without approximating exact distributions. Code is available at https://github.com/brushing-git/ModelsWithCause.

## 2 Related Work

Causal reasoning in NLP has emerged as a central research direction. While traditional NLP focused on predictive tasks, current research explores extracting causal relationships from text. Feder et al. (2021) demonstrate that causal modeling improves NLP robustness and interpretability through intervention estimation from text. Text-based confounder identification has enabled more robust causal inference (Keith et al., 2020). Causal mining systematically extracts causal knowledge from text corpora (Ali et al., 2021; Yang et al., 2022), with applications in social science research (Ash & Hansen, 2023).

Language models now directly perform causal inference and discovery on linguistic representations (Wang et al., 2024). Researchers evaluate LMs on causal reasoning benchmarks designed to test these capabilities (Liu et al., 2024), and they can guide conventional causal discovery algorithms (Ban et al., 2023). LMs can perform causal reasoning in natural language processing and chain-of-thought tasks (Jin et al., 2024) as evaluated in benchmarks such as CaLM Chen et al. (2024). However, LMs struggle with counterfactual reasoning (Yang et al., 2024; Ashwani et al., 2024) and are sensitive to word order when doing causal inference (Joshi et al., 2024). Proposed solutions include specialized causal models, multimodal approaches, and unified benchmarks, (Ma, 2024) while pre-trained LMs can enhance statistical causal discovery (Takayama et al., 2024), though commercial implementations they exhibit limitations (Kıcıman et al., 2023).

Evidence suggests LMs discover causal models from data: Li et al. (2024a) showed GPT develops internal game representations when predicting Othello moves. Nanda et al. (2023) identified simpler linear representations that capture these emergent structures. Transformers encode latent belief states in their residual streams that guide next-token predictions (Shai et al., 2024).

## 3 Preliminaries

Causal modeling operates on random variables as its core mathematical objects. Let $(\Omega_1, \Sigma_1, \Pr)$ be a probability space and $(\mathbb{R}^n, \mathfrak{B}(\mathbb{R}^n))$ be a Borel measurable space of real-valued vectors. An $n$-dimensional *random vector* on $(\Omega_1, \Sigma_1, \Pr)$ is a Borel measurable map $X : \Omega_1 \to \mathbb{R}^n$ with components $X_i = p_i \circ X$,

where $p_i : \mathbb{R}^n \to \mathbb{R}$ projects onto the $i$th coordinate. The probability function is $\Pr_X(X \in A) = \{\omega \in \Omega_1 : X(\omega) \in A\}$ for any $A \in \mathbb{R}^n$, with induced component probabilities $\Pr_{X_i} = \{\omega \in \Omega_1 : X_i(\omega) \in B\}$ for all $B \in \mathfrak{B}(\mathbb{R})$. The cumulative distribution function $F_X(\mathbf{x}) = \Pr_X(X_1 \leq x_1, \ldots, X_n \leq x_n)$ yields joint densities for continuous cases and mass functions for discrete cases. Causal discovery crucially relies on conditional independence relations between random variable sets. Two component sets $\mathbf{X}, \mathbf{Y}$ of random vector $X$ are conditionally independent given $\mathbf{Z}$ if and only if $\Pr_{\mathbf{X}}(\mathbf{X} \mid \mathbf{Z}, \mathbf{Y}) = \Pr_{\mathbf{X}}(\mathbf{X} \mid \mathbf{Z})$.

*Structural causal models* (SCMs) are defined over random variable sets as follows:

**Definition 3.1.** A structural causal model $M$ is a four-tuple $(\mathbf{U}, \mathbf{V}, \mathcal{F}, \Pr(u))$ where $\mathbf{U} = \{X_i \in X : i \in \{1, \ldots, n\}\}$ is a set of components from the random vector $X$ of length $n$ called the exogenous or unobserved variables, $\mathbf{V} = \{X_i \in X : X_i \notin \mathbf{U}\}$ is the complement of $\mathbf{U}$ in $X$ called the endogenous or observed variables, $\mathcal{F} = \{f_i : U_i \cup pa_i \to V_i : i = 1, \ldots, n\}$ where $U_i \subseteq \mathbf{U}$ and $pa_i \subseteq \mathbf{V} - V_i$ is the set of functions mapping from parents in both the unobserved and observed variables to each member $V_i \in \mathbf{V}$, $\Pr(u)$ is the joint distribution for the components of $X$ given in $\mathbf{U}$.

Each SCM partitions the random vector into observed and unobserved components, inducing a joint distribution over $\mathbf{V}$ through the unobserved distribution and functions in $\mathcal{F}$. The SCM induces a directed acyclic graph (DAG) $G$ with vertices $\mathbf{V}$. This yields a correspondence between graph properties and conditional independencies: vertices are *d-separated* conditional on a third vertex based on three fundamental structures (chain, fork, collider), and d-separation equivalently characterizes conditional independence (Pearl (2009), Theorem 1.2.4).

Under certain assumptions, conditional independencies reveal structural causal model properties. Classic examples include Pearl (2009)'s IC/IC* and Spirtes et al. (2001)'s PC/PC* algorithms. These algorithms identify the smallest SCM equivalence class when the generating model satisfies Markov, Minimality, and Faithfulness conditions[1]:

**Definition 3.2.** An SCM $M$ is *Markov* just in case the joint distribution of the observed component $\mathbf{V} = \{X_1, \ldots, X_m\}$ of our random vector for $m \leq n$ where $pa$ is the parents of $X_i$ in $M$:

$$\Pr_{\mathbf{V}}(X_1, \ldots, X_m) = \prod_{i=1}^{m} \Pr_{X_i}(X_i \mid pa(X_i)) \tag{1}$$

**Definition 3.3.** An SCM $M$ satisfies *minimality* if and only if for every proper subgraph $H$ of the SCM's DAG $G$ defined on the observed variables $\mathbf{V}$ with a vertex set given by $\mathbf{V}$, the corresponding set of variables in that graph is not Markov on $M$'s induced probability distribution.

**Definition 3.4.** An SCM $M$ is *faithful* if its DAG $G$ is such that any set of observed variables $\mathbf{X}, \mathbf{Y}, \mathbf{Z}$, the corresponding vertices for $\mathbf{X}$ is d-separated from $\mathbf{Y}$ by $\mathbf{Z}$ if and only if $\mathbf{X}$ is conditionally independent of $\mathbf{Y}$ given $\mathbf{Z}$ in $M$ induced distribution Pr.

These conditions, plus *causal sufficiency* (no unobserved common causes) for some algorithms, enable recovery of the minimal partial DAG set (Pearl, 2009; Spirtes et al., 2001) (we discuss how these assumptions put limitations on our methods in section 7).

IC and PC algorithms have two limitations. First, they require exponential worst-case time for skeleton discovery (Le et al., 2016).[2] Second, they return only an equivalence class of DAGs that could have generated the data. Thus, these algorithms *underdetermine* the true causal model. Generally, probabilistic information alone cannot identify interventional relationships (Bareinboim et al., 2022). Additional assumptions are required for unique causal model identification.

---

[1] While PC applications typically assume i.i.d. data (Glymour et al., 2019; Hasan et al., 2023), the algorithm only requires consistent statistical tests for distinguishing dependence from independence (Spirtes et al., 2001). Extensions exist for non-i.i.d. temporal data (Niu et al., 2024).

[2] Skeleton discovery in Bayes networks is NP-complete (Chickering, 1996).

**Algorithm 1** Algorithm for causal discovery.

1: Apply PC or IC discovery algorithm
2: **for all** undirected edges between variables $v_i, v_j$ **do**
3:     **if** $i < j$ **then**
4:         Direct edge from $v_i$ to $v_j$
5:     **else**
6:         Direct edge from $v_j$ to $v_i$
7:     **end if**
8: **end for**

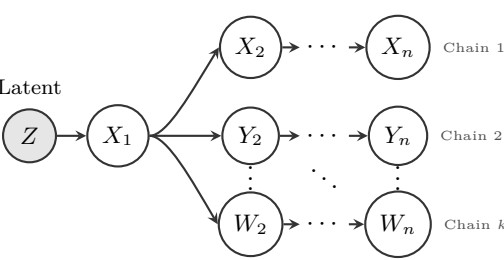

Figure 1: (Left) Algorithm for causal discovery. (Right) Causal model for Markov chain generation with $k$ possible chains. The latent variable $Z$ determines chain selection through initial token $X_1$. From $X_1$, the model generates $k$ different sequences following distinct Markov dynamics, with vertical ellipses indicating chains 3 through $k - 1$.

## 4   A Theoretical Reason for Why LMs Can Do Causal Discovery

Completing a partial DAG requires orienting its undirected edges. This requires determining the correct interventional factorization: for neighbors $X_1$ and $X_2$ with an undirected edge, we must identify whether $\Pr(X_1|do(X_2 = x)) = \Pr(X_1|X_2 = x)$ or $\Pr(X_1|do(X_2 = x)) = \Pr(X_1)$. Bareinboim et al. (2022) prove this is impossible from observational data alone (theorem 27.1). Additional assumptions enable causal discovery. Temporal ordering, if available, uniquely identifies the DAG structure of the generating SCM.

**Definition 4.1.** Let $M = (\mathbf{U}, \mathbf{V}, \mathcal{F}, \Pr(u))$ be a SCM. A temporal ordering of $(v_0, v_1, \ldots, v_n)$ on $V$ is a total ordering such that for any $i < j$, $v_i$ does not occur after $v_j$ in time.

Algorithm 1 leverages temporal ordering to orient all edges in the CPDAG (Pearl (2009), chapter 3). This algorithm recovers the true DAG $G$ under two conditions: causes precede effects temporally (Pearl (2009), definition 2.7.4) and unobserved variables have at most one child. This can be demonstrated in a straightforward proposition.

**Proposition 4.2.** *Let $M = (\mathbf{U}, \mathbf{V}, \mathcal{F}, \Pr(u))$ be a SCM with the corresponding DAG $G$ of observed variables $V$ and induced probability function $\Pr$. Suppose $M$ satisfies the Markov, faithfulness, and temporal ordering conditions and that for each $U \in \mathbf{U}$ has at most one child. Then there exists an algorithm for finding $G$.*

Thus, appropriate assumptions enable unique recovery of causal directions (as was suggested by Spirtes et al. (2001, p.93) and is well known). The proof is in appendix A. Two things should be noted. First, these minimal conditions may require augmented PC algorithms, for instance, require i.i.d. data for valid conditional independence tests. Second, real data may violate our assumptions through non-Markov processes, faithfulness violations, unobserved confounders, or missing temporal information.

Given this theoretical foundation, causal discovery requires two inductive biases: identifying appropriate probabilistic properties, e.g. conditional independencies and recognizing temporal order. Do LMs possess these biases?

LMs can identify temporal information through positional encodings, which transformers require due to their permutation invariance (Vaswani, 2017; Dufter et al., 2022). Absolute sinusoidal positional encoding adds the following to each sequence element's embedding (with dimension $d$):

$$\text{PE}_{pos,k} = \begin{cases} \sin\left(\frac{pos}{10000^{k/d_{\text{model}}}}\right) & \text{if } k \text{ is even} \\ \cos\left(\frac{pos}{10000^{(k-1)/d_{\text{model}}}}\right) & \text{if } k \text{ is odd} \end{cases} \tag{2}$$

This enables transformers to track input order, thereby capturing temporal structure when present in the data.

Can LMs identify the probabilistic properties required for causal discovery? Lu & Lu (2020) demonstrate that ReLU networks can approximate arbitrary probability distributions to any Wasserstein-1 distance in the data limit. However, universal approximation does not guarantee that gradient descent finds these approximations on finite data (except potentially when overparameterized with infinite data (Rotskoff & Vanden-Eijnden, 2018)). Empirical testing must verify whether transformers learn the requisite probabilistic properties.

We test whether generative models learn correct conditional independencies. However, our experiments reveal that conditional independence alone insufficiently evaluates LMs' causal discovery capabilities. Probabilistic symmetry properties provide an alternative evaluation. In the Bayesian interpretation, probabilities represent an agent's degrees of belief rather than external facts. Causal discovery then becomes identifying which mixture of causal models the agent's probabilities encode. De Finetti's theorem states that exchangeable (permutation-invariant) Bernoulli sequences admit representation as mixtures over i.i.d. distributions conditioned on a latent variable (Kirsch, 2019). Importantly, pre-trained LMs appear to demonstrate these symmetry properties in-context (Zhang et al., 2023; Falck et al., 2025). If LMs learn exchangeability, causal discovery reduces to identifying the correct mixture component—effectively detecting conditional independencies without explicit latent variable search.

Exchangeability generalizes to Markov exchangeability. The *transition count matrix* for a discrete random vector with $n$ states records transitions between values $i = 0, 1, \ldots, n$:

$$\begin{pmatrix} a_{00} & a_{01} & \ldots & a_{0n} \\ a_{10} & a_{11} & \ldots & a_{1n} \\ \vdots & \vdots & \ddots & \vdots \\ a_{n0} & a_{n1} & \ldots & a_{nn} \end{pmatrix} \tag{3}$$

where each $a_{ij}$ is the transition counts from values $i$ to $j$. We can then define Markov exchangeability:

**Definition 4.3.** Let $\Pr_X$ be a probability function defined on the random vector $X = [X_1, \ldots, X_n]$ and let $\rho \mapsto j \neq i$ for $i, j \in \mathbb{N}/\{1\}$ be any permutation that maps the $i$-th component to the $j$-th component such that the resulting random vector has the same 1-st component and transition matrix as $X$. $\Pr_X$ is *Markov exchangeable* if:

$$\Pr_X(X_{\rho(1)} = x_1, \ldots, X_{\rho(n)} = x_n) = \Pr_X(X_1 = x_1, \ldots, X_n = x_n) \tag{4}$$

For recurrent processes (where some state $x \in S$ satisfies $\Pr_X(X_i = x$ infinitely often$) = 1$), Diaconis & Freedman (1980) prove that Markov exchangeable distributions are *uniquely* representable as Markov chain mixtures (theorem 22).

Therefore, we can use the symmetry property of Markov exchangeability as an additional test to identify when probabilistic models have learned *some* mixture of Markov chains. It is an orthogonal property to conditional independence that can be used to identify when LM probabilities encode the right kind of graphical structure without specifying the exact Markov processes involved.

## 5 Experiments

In this section, we document our experimental methods and results. We describe our synthetic datasets and model architectures, then report results for three experiments: conditional independence tests, probabilistic symmetry tests, and distribution approximation analyses.

| | $X_{i-1} = 1$ | $X_{i-1} = 2$ | $X_{i-1} = 3$ | $X_{i-1} = 4$ | $X_{i-1} = 5$ | $X_{i-1} = 6$ |
|---|---|---|---|---|---|---|
| $\chi^2$ **P-value** | 0.028 | 0.023 | <0.001 | 0.025 | 0.003 | <0.001 |
| **Observed Cramer's V** | 0.067 | 0.051 | 0.033 | 0.065 | 0.046 | 0.030 |

Table 1: Baseline Chi-Squared Tests: Columns indicate $X_{i-1} = k$ for $k \in \{1, \dots, 6\}$. Tests check whether the $i$-th variable is conditionally independent of all $j > i-1$ variables given the $(i-1)$-th and 0-th variables. Rows show average p-values (null: variables dependent, threshold $V = 0.15$) and observed Cramer's V. Results from $N = 100{,}000$ i.i.d. sequences; all standard deviations $\approx 0.00$.

## 5.1 Experimental Methods and Procedures

We use synthetic data for two reasons: real-world datasets lack known generating models, and synthetic data provides a best-case scenario for testing language models. Synthetic data provides three advantages. It allows experimental control over the generating distribution, precise evaluation through direct comparison with ground truth, and generalizability to any discrete sequential dataset. We use a mixture-of-Markov processes, one of the simplest causal models satisfying both proposition 4.2 and the Markov exchangeability conditions from definition 4.3. Figure 1 shows this model's causal structure.

Each dataset consists of sequences with categorical values $i \in \{0, 1, 2, 3, 4, 5\}$. We generate each sequence by first sampling from a prior distribution $\Pr(X_1)$, which determines the stochastic matrix governing subsequent tokens (see appendix B for examples). Once set, we sample from this matrix to generate the remaining tokens and record each sequence's log probability. This makes the generating process a mixture of Markov chains as given in figure 1. We test sequence lengths of 6, 100, and 500 tokens, training separate models for each length. We report results for 100-token sequences; results for 6- and 500-token sequences were identical. Dataset sizes ranged from 62,500 to 6,250,000 sequences, selected to match Chinchilla-optimal token counts for our model parameters (Hoffmann et al., 2022). We split all datasets 80/20 into training and validation sets. By construction, the data distribution is Markov, minimal, and faithful.

We evaluate four architectures: Neural Autoregressive Distribution Estimator (NADE) (Uria et al., 2016), Encoder-Decoder transformer (Vaswani, 2017), Decoder-only transformer (Radford, 2018), and Switch transformer (Fedus et al., 2022; Shazeer et al., 2017). Appendix C details architecture specifications and selection criteria. For trained models in our experiments, we trained on five different random seeds, which included different weight initializations, dataset train/test splits, and randomized batches, and averaged the results across models of the same architecture. After hyperparameter selection and training, we conducted the experiments described below.

## 5.2 Identifying Conditional Independence

Our first experiment tests whether models learn the data's conditional independence structure. Since we generated each dataset via a component in a mixture of Markov processes, each token's conditional probability depends only on its immediate predecessor and the initial token, which implies equality between that conditional probability and the token conditional on all tokens that came previously. We test whether models learn this conditional independence:

$$\Pr_m(X_i = x_i \mid X_{i-1} = x_{i-1}, \dots, X_1 = x_1) = \Pr_m(X_i = x_i \mid X_{i-1} = x_{i-1}, X_1 = x_1) \tag{5}$$

As a baseline for checking whether PC would identify the mixture-of-Markov chains structure, we run $\chi^2$ equivalence tests on randomly drawn sequences from a training dataset. This checks for conditional independence against the equivalence null that they are dependent. Table 1 reports the average p-values and observed Cramer's V. We find that sequences are conditionally independent as warranted by equation 5.

To assess whether models learn this Markov property, we test the conditional independence in Equation 5. We measured the similarity between these distributions using the Jensen-Shannon Divergence (JSD); JSD

| Model | JSD-Train 0.95 CI | JSD-NT $\mu$ |
|---|---|---|
| NADE | (0.012, 0.013) | 0.002 |
| Enc.-Dec.-T. | (0.143, 0.147) | 0.018 |
| Decoder-T. | (0.297, 0.301) | 0.057 |
| Switch-T. | (0.261, 0.265) | 0.051 |

(a) Markov Property Tests: The Jensen-Shannon Divergence (JSD) assesses if models learn the Markov property as defined in Eq. 5; a value closer to 0 indicates a better fit. The table shows 95% confidence intervals (CI) for the average of 5 trained models and the mean JSD for 5 non-trained models. JSD tests used $N = 6388$.

| Model | Train CI $\mu$ | No-train CI $\mu$ |
|---|---|---|
| Enc.-Dec.-T. | (3.00, 3.14) | (-5.75, -5.59) |
| Decoder-T. | (3.19, 3.32) | (-6.21, -6.02) |
| Switch-T. | (3.98, 4.13) | (-5.60, -5.42) |

(b) Log Probability Differences: The mean log probability difference between a model's assigned probability and the true probability for a sequence for 1 model on fixed random seed; a value closer to 0 indicates better distribution approximation. The table shows 95% confidence intervals (CI) for both trained and non-trained models. Log probability tests used $N = 1598$.

Table 2: Comparison of model performance metrics.

ranges from 0 (identical distributions) to 1 (maximally difference).[3] For each position $i$ in a sequence, we obtain two distributions from the model $m$ through separate forward passes. The first uses full context $X_{i-1}, \ldots, X_1$ to predict $X_i$; the second uses only $X_{i-1}, X_1$. We then compute the JSD between these distributions. We test both trained and randomly initialized models.

Table 2a shows these results. The 95% confidence intervals for trained transformers' mean JSD are close to but above zero. An independent-samples t-test (80% power) confirms that trained models' JSD significantly differs from random initialization. This indicates that training induces partial learning of conditional independence structure, though not perfect adherence to the Markov property. Furthermore, we observe that non-trained model exhibit near-0 JSD means. This occurs because the random Xavier initialization produces weight distributions that render the model's predictions approximately independent of the input context. Since both conditional distributions $\Pr_m(X_i \mid X_{i-1}, X_1)$ and $\Pr_m(X_i \mid X_{i-1}, \ldots, X_1)$ are approximately independent of their conditioning sets, they trivially match each other, which yields low JSD. We therefore conduct additional symmetry tests (section 5.3) to better characterize what models learn.

### 5.3 Symmetry Property Experiments

Since conditional independence tests cannot discriminate trained from untrained models, we test Markov exchangeability to determine whether models learn the mixture-of-Markov-processes structure. We measure the probability ratio between a sequence $X_1, \ldots, X_n$ and its Markov-exchangeable permutation $X_{\rho(1)}, \ldots, X_{\rho(n)}$. Markov-exchangeable permutations are generated from the transition count matrix of a sequence: we perform a backtrack search over sequences that conform to the transition counts of the original sequence we aim to permute. Assigning equal probabilities between sequences with the same starting sequence value and transition counts implies the model probability function has identified the generating mixture Markov chain. Successful learning of Markov exchangeability across all chains implies learning of the full generating mixture model. For numerical stability, we use log probabilities, converting the ratio to a difference: $\log \Pr_m(X_1, \ldots, X_n) - \log \Pr_m(X_{\rho(1)}, \ldots, X_{\rho(n)})$. Values near 0 indicate nearly identical probabilities. We compute model joint probabilities using the autoregressive factorization: $\log \Pr_m(X_1, \ldots, X_n) = \sum_{i=1}^{n} \log \Pr_m(X_i \mid X_1, \ldots, X_{i-1})$ where we obtain each conditional probability from the model's softmax output for position $i$ with the $1, \ldots, i-1$ input context. We hypothesize that trained models exhibit two properties: (1) mean log differences near 0 for Markov-exchangeable permutations, and (2) variance below 1.0 (indicating concentration). These properties should fail for random sequence pairs. We conjecture that training induces these properties, which randomly initialized models lack.

Our experimental procedure consists of five steps: (1) sample $N = 1598$ sequences from the train/test split; (2) for each sequence, generate 3 Markov-exchangeable permutations and select 3 random sequences from both the training and test distribution; (3) compute model log probabilities for all sequences, permuta-

---

[3]Our conditional independence test does not require i.i.d. data. Both our generated data and the generative models learning from it are explicitly non-i.i.d. Traditional tests would fail in this setting, but the causal discovery requirements from Proposition 4.2 are still satisfied when a model can learn the necessary conditional independence structures.

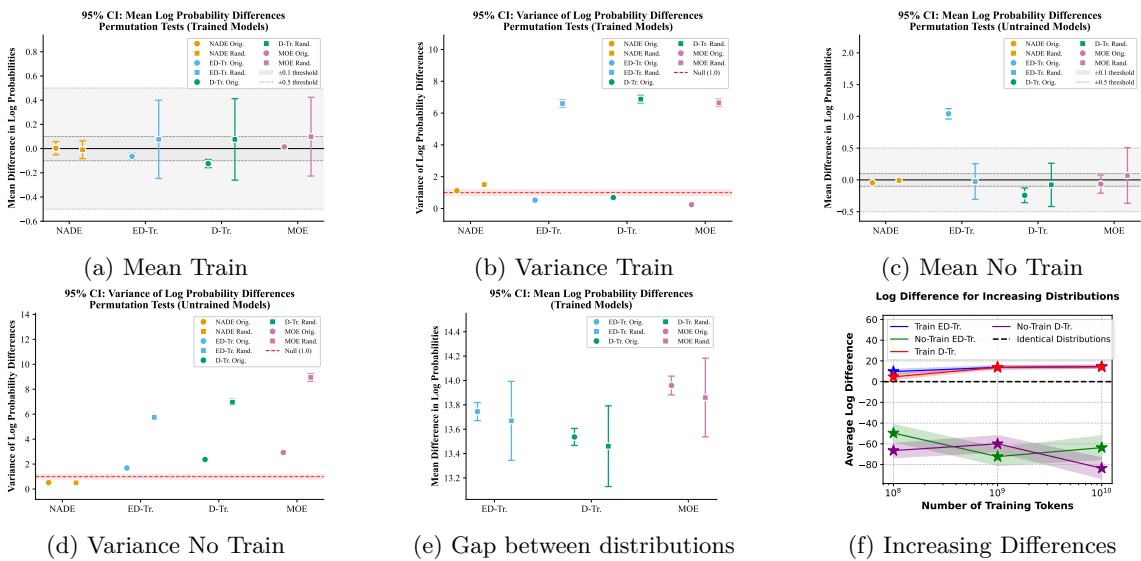

Figure 2: 95% CI for: (a-b) Mean and variance of log differences between sequences and permutations; (c-d) Comparison between trained vs. non-trained models; (e) Gap between model and true probability assignments; (f) Impact of training data volume on model confidence. Dashed lines and shaded areas indicate null hypothesis thresholds. Trained model results are averaged across 5 models trained on different random seeds.

tions, and random samples; (4) calculate log probability differences between the target sequence and its permutations and random samples; (5) average differences across permutations and random samples.

We conduct frequentist and Bayesian equivalence tests with bounds $[x_{lower}, x_{higher}] \in \{[-1.0, 1.0], [-0.5, 0.5], [-0.1, 0.1]\}$. We reject the null hypotheses $\mu_0 > x_{higher}$ or $\mu_0 < x_{lower}$ when the mean falls within the bounds. For variance, we test $\sigma_0^2 > 1.0$, where rejection indicates concentration around 0. We compute 95% confidence intervals and Bayesian credible/highest density intervals (details in appendix D). Figure 2 shows the results. All models except the Decoder-only Transformer reject the null hypothesis at all thresholds (circles in Figure 2a). However, comparing sequences to random samples fails to reject the null at the $[-0.5, 0.5]$ threshold (squares in Figure 2a), though they fail to reject the null at the tightest test. For variance tests, all models except NADE reject the null for Markov-exchangeable permutations (circles in Figure 2b) but not for random sequences (squares), with large effect sizes. These results indicate that Encoder-Decoder and Switch Transformers learn Markov exchangeability, the Decoder-only Transformer nearly learns it, and NADE does not.

Our second hypothesis posits that training induces Markov exchangeability rather than it emerging from random initialization of the architecture alone. We compare trained and randomly initialized models. All non-trained models were initialized as is standard with Xavier distributions on the weights. Figure 2 presents these comparisons. For randomly initialized models (Figure 2c), NADE shows results identical to trained models, while other architectures fail the strictest null hypothesis test ($[-0.1, 0.1]$), which provides weak evidence that non-NADE architectures exhibit some inherent Markov exchangeability. Variance analysis reveals stark differences (Figure 2d): randomly initialized models (except NADE) show significantly higher variances that fail to reject the null hypothesis. This confirms that training induces partial learning of the mixture-of-Markov-chains structure for all models except NADE, consistent with the conditional independence results (section 5.2).

## 5.4 Testing Longer Range Dependencies

Our experiments provide credible evidence that common LM architectures can learn the probabilistic properties necessary to divine simple first-order Markov chain SCMs. To test whether they can learn longer

range dependencies, we conducted the same battery of tests on a single traditional Transformer architecture trained on mixtures of second-order Markov chains. The factorization from each component of the generated process is given by $\Pr_{dist}(X_1, \ldots, X_n) = \Pr(X_1, X_2) \prod_{i=3}^{n} \Pr(X_i \mid X_{i-1}, X_{i-2})$. Our tests for Markov exchangeability are identical as before except we select valid permutations based on a two-block transition count matrix. The results are given in appendix E . We find the same pattern of previous results: trained models show tight confidence intervals around a log mean difference of 0 and below 1 in variance while random sequences and non-trained models violate one or the other. These results suggest models can learn SCMs with longer-range dependencies.

## 5.5 Learning by Approximating the Underlying Distribution

One hypothesis for the observed learning is that models approximate the true data distribution, correctly estimating sequence probabilities. Universal approximation theorems for feedforward networks with ReLU activations (Lu & Lu, 2020) support this possibility.

Our synthetic data enables exact computation of each sequence's $X_1, \ldots, X_n$ true probability from the prior and transition matrices. We measure distribution approximation as the log probability difference between the model's distribution and the true distribution: $\log \Pr_m(X_1, \ldots, X_n) - \log \Pr_{dist}(X_1, \ldots, X_n)$. As a baseline, we also compare model probabilities for random sequences against the true probabilities of $X$. We apply the same statistical tests as in section 5.3. We also compute Wasserstein-1 distances, but all values are near zero, including for random sequences. We therefore omit these uninformative results.

Figure 2e shows 95% confidence intervals for mean differences across $N = 1598$ samples, relative to our null hypotheses. All architectures assign significantly higher probabilities than ground truth—not only to original sequences but also to random sequences, leading us to reject the distribution approximation hypothesis.

We next investigate whether overconfidence stems from insufficient data exposure. Training data represents a tiny fraction of the space; the overall number of possible sequences that could be observed ($6.25 \times 10^7$ training sequence compared to $6.53 \times 10^{77}$ possible sequences). We conduct two experiments to test this. First, we vary training data size across multiple orders of magnitude. If limited data causes overconfidence, models should assign probabilities closer to ground truth as dataset size increases. Memory constraints limit this experiment to Encoder-Decoder and Decoder-only transformers ($N = 1598$ sequences). Second, we create a length-6 dataset containing 46,656 possible sequences and densely sample 500,000 training sequences. With exposure to most possible sequences, models should memorize exact probabilities if overconfidence results from limited data.

Figure 2f shows the first experiment's results. Counterintuitively, log probability gaps *increase* with more training data even as the relative order of magnitude of the number of tokens increases. This means that the models grow more confident about the probabilities as they see more data. To verify this is not an architectural artifact, we examine randomly initialized models. Untrained models show the opposite pattern of slightly decreasing confidence as dataset size increases, though we believe the decrease is an empirical artifact.

Table 2b presents the second experiment's results. Even with near-complete data coverage, all models assign probabilities significantly above ground truth, exhibiting systematic overconfidence. This definitively rules out distribution approximation as the learning mechanism.

Given that models do not learn exact probabilities, we test whether they learn *qualitative rankings*. We examine how models rank token probabilities at each position in 100-token sequences. For $N = 1598$ sequences, we compare model rankings to the true transition matrix. Table 3 shows mean similarity scores. We compute two metrics: a pairwise ranking similarity (appendix D) and Levenshtein distance with substitution costs. Trained models show substantially higher ranking similarity than both random baselines and untrained models, except for the Decoder-only Transformer. This shows that the models learn ordinal structure—correct relative rankings—while failing to learn cardinal structure, i.e. correct absolute probabilities.

| Model | Similarity T | Similarity NT | Levenshtein T | Levenshtein NT |
|---|---|---|---|---|
| Enc.-Dec.-T. | 0.55 ($\pm$ 0.06) | 0.48 ($\pm$ 0.06) | 0.31 ($\pm$ 0.04) | 0.23 ($\pm$ 0.05) |
| Decoder-T. | 0.55 ($\pm$ 0.06) | 0.51 ($\pm$ 0.03) | 0.31 ($\pm$ 0.04) | 0.27 ($\pm$ 0.03) |
| Switch.-T | 0.55 ($\pm$ 0.06) | 0.50 ($\pm$ 0.06) | 0.31 ($\pm$ 0.04) | 0.25 ($\pm$ 0.04) |
| Random Assignment | 0.02 ($\pm$ 0.00) | 0.02 ($\pm$ 0.00) | 0.01 ($\pm$ 0.00) | 0.01 ($\pm$ 0.00) |

Table 3: Similarity metrics between model probability rankings and ground truth (T=Trained, NT=Non-Trained). Higher values indicate greater similarity to transition matrices. Trained model rankings were averaged across 5 trained models. All models significantly outperform random baselines.

## 6  Discussion

Prior work on causal discovery with LMs utilizes LMs to extract causal relationships *represented* in text, such as identifying whether smoking causes cancer from medical literature. Similarly, applications of LMs to causal inference estimate causal effects using token-level representations of observed variables. We address a more fundamental question: can LMs discover the causal processes that *generate* token sequences?

Token-level causal learning could support semantic understanding in at least two cases. First, when causal relationships in text are consistently expressed temporally in text, e.g. "X happened, then Y, which caused Z". Semantic causal relationships can then be extracted from token causal relationships. Second, the token generation process reflects the author's causal model of the domain, e.g. there are latent variables explaining the sequential ordering such as the transition matrices used in our experiments. Our results show that LMs possess the necessary inductive biases for token-level causal discovery. Verifying whether both of these cases are viable for semantic understanding remains an open empirical question.

Understanding LM inductive biases for causal discovery is critical because traditional methods scale poorly to high-dimensional token spaces. Constraint-based methods (e.g., PC, FCI), score-based methods (e.g., GES), and functional causal models (e.g., LiNGAM) offer sound theoretical guarantees but scale poorly with dimensionality. In contrast, LMs learn compressed representations of high-dimensional distributions, enabling tractable causal discovery when their inductive biases align with causal structure. LMs offer an additional advantage: while traditional methods require i.i.d. observations, LMs require only that training sequences are sampled i.i.d., not that tokens are independent. This relaxed assumption enables causal discovery on sequential data with inherently dependent observations, such as natural language corpora.

Our experiments demonstrate that LMs possess appropriate inductive biases for causal discovery on temporally ordered data. Theoretically, Proposition 4.2 establishes that unique causal discovery on temporal data requires both identifying probabilistic symmetries and leveraging temporal order. Positional encoders, standard in LM architectures, naturally provide temporal ordering information. Our experiments show that LMs learn to identify these probabilistic symmetries. Models identify these symmetries through ordinal probability rankings rather than accurate distributional approximation. We hypothesize this behavior stems from systematic overconfidence in transformer architectures and training procedures, a well-documented phenomenon (Guo et al., 2017; Wei et al., 2022). While LMs show promise for causal discovery, their capacity for accurate causal effect estimation may be severely limited.

Our results inform ongoing debates about whether LMs can perform causal reasoning. These debates focus on reasoning about causal variables *represented* or *described* in text. A prerequisite question is whether LMs can learn representations of the causal processes generating their training data. While initial work suggested LMs learn coherent world models (Li et al., 2024a), subsequent experiments found they learn incoherent physical rules (Vafa et al., 2025). This literature could benefit from applying formal causal models from causal inference and discovery. Our experiments suggest that LMs can learn representations by discovering the causal processes generating their training data. However, systematic overconfidence may bias LMs toward learning only ordinal rankings, indicating reliance on heuristics rather than principled probabilistic inference. Improved calibration to the true data-generating distribution may therefore reduce this bias toward improper representations.

## 7   Limitations

Our conclusions have several important limitations.

First, our experiments address identifying the processes that generate tokens, not extracting causal relationships described in text. We provide no evidence for LMs' ability to discover causal models described in text, such as the effect of drug interventions on disease outcomes, or to leverage pre-existing human knowledge about causal relationships described in text.

Second, our results assume no unobserved confounders. The causal sufficiency requirement is most likely to be violated in practice. Tokens can be generated by unobserved variables, such as grammatical rules, that confound observed statistical relationships. In these cases, even traditional algorithms adapted for them, like FCI, are unable to recover a minimal Markov equivalence class. Nevertheless, they permit the discovery of some causal relationships as well as identifying potential hidden confounders. We leave to future work whether LMs can be successfully used on data with latent confounders.

Third, our experiments tested model distributions and not representations. We showed those distributions combined with the positional encoder show the right inductive biases for learning simple SCMs. We did not show what single or multiple representations (Marshall & Kirchner, 2024) the model might have of those SCMs. Future work would need to be done to extract those representations and whether they are produced by an isomorphic SCM.

Fourth, most traditional causal discovery applications involve unordered tabular data without natural temporal structure. Whether LMs retain these inductive biases for non-sequential data remains open.

Fifth, while LMs learn qualitative probability rankings sufficient for causal discovery, this may impair causal effect estimation. Ordinal rankings are insufficient for identifying probability magnitudes; an LM may identify one probability as greater than another but fail to map how much greater than another. Effect estimation requires a comparison of probabilistic magnitudes. Therefore, better calibrated model distribution estimates would improve the causal effect estimation.

Sixth, our experiments use synthetic data from first-order and second-order Markov processes rather than naturalistic text corpora. Validating these findings on natural language data is essential future work. Two potential targets would include causal discovery with respect to unobserved or latent variables, such as grammatical rules, and work on mathematical and coding benchmarks, where causal relationships might exist.

### Acknowledgments

We are appreciative of the feedback received in the drafting of this paper, especially from audience members at the Society for Philosophy and Psychology, from anonymous reviewers, and from Daniel Herrmann and Ben Levinstein. Funding support for this work was provided by Purdue University, the Ross Lynn Fellowship, the College of Liberal Arts' VRAI Lab, and University of Virginia Research Computing.

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

## A    Proof of Proposition 4.2

We claim the algorithm in 1 will produce $G$. Since $M$ is Markov and faithful, then the PC or IC algorithm will produce the Markov equivalence class of $G$, which is represented in the completed CPDAG, $H$. We then show that applying the algorithm will produce $G$ by induction.

Base case: consider all neighbors of $v_0$. Any edges involving $v_0$ must be directed away since in $H$, any neighbors will be undirected or away due to colliders, while any additional undirected neighbors will have been oriented away due to the temporal ordering, which respects causation and so will be respected in $G$.

Inductive case: suppose that for variables $(v_0, \ldots, v_{n-1})$, the edges are oriented correctly in as in $G$. Consider variable $v_n$. For any neighbor $v_i$, either $i < n$ or $i > n$. If $i < n$, then by hypothesis the edges are oriented correctly in $G$. If $i > n$, then the algorithm will have oriented them from $v_n$ to $v_i$, which will be respected in $G$ since $v_n$ occurs before $v_i$.

## B  Example Priors and Stochastic Matrices

| Starting Token | Prior |
|:---:|:---:|
| 0 | 0.061742605 |
| 1 | 0.10922861 |
| 2 | 0.287429074 |
| 3 | 0.063535748 |
| 4 | 0.141853226 |
| 5 | 0.336210737 |

Table 4: Sample priors for the 100-length dataset used on the Encoder-Decoder Transformer. Prior probabilities were taken from a normalized uniform distribution $\mathcal{U}(0,1)$. Each probability corresponds to the probability of the first token in the sequence, which then has a corresponding stochastic matrix.

| | 0 | 1 | 2 | 3 | 4 | 5 |
|:---:|:---:|:---:|:---:|:---:|:---:|:---:|
| 0 | 0.234312928 | 0.27023688 | 0.00602517 | 0.22205389 | 0.207990725 | 0.059380408 |
| 1 | 0.054615949 | 0.248166706 | 0.113736672 | 0.246896763 | 0.197805355 | 0.138778555 |
| 2 | 0.322262131 | 0.293188673 | 0.12449644 | 0.01518472 | 0.106163398 | 0.138704637 |
| 3 | 0.17178788 | 0.242553914 | 0.086731108 | 0.185821451 | 0.144548306 | 0.168557341 |
| 4 | 0.065308119 | 0.172370142 | 0.104083311 | 0.148420859 | 0.221742868 | 0.288074701 |
| 5 | 0.040421359 | 0.04995777 | 0.12288811 | 0.252494683 | 0.323080174 | 0.211157903 |

Table 5: Sample recurrent stochastic matrix for the prior token 2 in table 4. This matrix was used to produce all of the sequences that the Encoder-Decoder Transformer architecture was trained and tested on. The matrix was generated by sampling each row from a uniform probability distribution $\mathcal{U}(0,1)$ and then renormalized.

## C  Details on Training and Architectures

For selecting our model architecture, we performed pairwise grid search on the following parameters with 10-fold cross-validation on a small dataset of 60000 100-length sequences trained on each fold for 25 epochs. For learning rates, we tested $[0.1, 0.01, 0.001, 0.0001, 0.00001]$; for the NADE, we tested $[8, 16, 32, 64, 128, 256, 512, 1024, 2048, 4096]$ hidden units; for the various transformer architectures, we tested $[2, 5, 10]$ heads, $[2, 4, 6, 8, 10, 12]$ encoder and decoder layers, dropout probabilities of $[0.0, 0.1, 0.2, 0.3, 0.4]$, feedforward networks with $[8, 16, 32, 64, 128, 256, 512, 1024, 2048, 4096]$ hidden units; for the switch transformer, we also tested $[2, 3, 4, 5, 6]$ experts with the top 2 experts. The corresponding NADE, Encoder-Decoder transformer, Decoder-only transformer, and Switch transformer architectures found to have the lowest cross-validation losses are given in table 6. Note that for the Switch transformer, we used a decoder-only Switch transformer. For our short sequence tests, we used sequences of length 6. This required us to revise our architectures slightly by changing the number of heads to 6 for all architectures.

| Model | LR | FF | Heads | Enc. Lyrs | Dec. Lyrs | Dropout | Experts |
|:---:|:---:|:---:|:---:|:---:|:---:|:---:|:---:|
| NADE | 0.0001 | 16 | NA | NA | NA | NA | NA |
| Transformer | 0.0001 | 2048 | 5 | 2 | 8 | 0.0 | NA |
| Dec. Trans. | 0.0001 | 4096 | 10 | NA | 4 | 0.0 | NA |
| Switch Trans. | 0.0001 | 4096 | 10 | NA | 4 | 0.0 | 5 |

Table 6: The various architectures we used in our primary training experiments, selected via grid search and 10-fold cross-validation.

Once selected, all models were trained on their respective training datasets using the Adam optimizer with a cosine annealing learning rate scheduler for the transformers and a step scheduler for the NADE. The NADE was trained for 250 epochs with each mini-batch being 100 samples, with the learning rate halved every 50 epochs. The Encoder-Decoder and Decoder-only transformers were trained for 4 epochs with each mini-batch being 250 samples; the Switch transformer was trained for only 2 epochs with each mini-batch being 100 samples. Models were assessed on validation set drawn 80/20 from the training distribution. We observed during training that all models showed continual decreases in training and validation loss.

## D Statistical Methods

For our symmetry experiments, we ran equivalence tests and computed confidence intervals, credibility intervals, and highest density intervals for the averages. Our equivalence tests consisted of two, one-sided t-tests to define our null hypotheses. Our alternatives were $(-1.0, 1.0)$, $(-0.5, 0.5)$, $(-0.25, 0.25)$, and $(-0.1, 0.1)$, which correspond to the models' having between 0.36 times smaller or 1.7 times greater probabilities, 0.61 times smaller or 0.64 times greater probabilities, 0.78 times smaller or 0.28 times greater probabilities, 0.90 times smaller or 0.10 times greater probabilities assigned to the original sequence compared to Markov exchangeable permutations. This effectively bounds the relative ratio of probabilities within $2, 1, 0.5$, and $0.2$. Each one-sided t-test checked the null hypothesis below or above the bound, depending on the side. For the variances, we used a one-side t-test with the null $\sigma^2 > 1.0$ and the alternative being $\sigma^2 \leq 1.0$. We ran a chi-square test against the null and computing the 0.95 confidence intervals. Credibility and highest density intervals were computed concerning the prior $\mathcal{N}(0, 10)$, which we chose to ensure a good enough spread of possible values outside of the standard normal range. All tests unless otherwise noted were run with $N = 1598$ samples for 80% power for an effect size of 0.1. We estimated the population variance on a trial dataset, found it to be roughly 1, and then computed at $\beta = 0.2$ with an effect size of 0.1 our necessary sample size.

For the qualitative rankings we use a pair-wise similarity metric and Levenshtein distance with substitution cost at unit. Our similarity metric works by first placing a descending order on token's in terms of their log probabilities. Higher in the order means a token has higher probability relative to its peers for that place. We acquire these probabilities directly from a log softmax over the model's logits and from the corresponding row in the correct stochastic matrix. After ranking the tokens, we then start with the model's rankings $R_m$ and proceed down the rank. At each token $t_j$, we take the set of all tokens lower in the ranking $T_m = \{t_i \in R_m : i < j\}$ and intersect that with all tokens lower than $t_j$ in the stochastic matrix ranking $R_d$, $T_d = \{t_i \in R_d : i < j\}$. We use the cardinality of this set and divide it by the cardinality of the set $R_m$ and this provides our pairwise similarity measure for token $t_j$:

$$s(t_i) = \begin{cases} \frac{|T_m \cap T_d|}{|T_m|} & \text{if } |T_m| > 0 \\ 0 & \text{otherwise} \end{cases} \tag{6}$$

The similarity $S$ of the two rankings $R_m, R_d$ with $k$ tokens is the sum over these pairwise rankings divided by the $k$:

$$S(R_m, R_d) = \frac{1}{k} \sum_{j=1}^{k} s(t_j) \tag{7}$$

Finally, the similarity for the whole sequence of length $n$ is simply the sum of the rankings for each member of the sequence divided by $n$.

| Model | JSD-Train $\mu$ | JSD-No-train $\mu$ | T-Statistic | p-value | 0.95 CI |
|---|---|---|---|---|---|
| NADE | 0.015 | 0.002 | 97.594 | 0.0 | (0.004, 0.031) |
| Transformer | 0.156 | 0.037 | 325.517 | 0.0 | (0.072, 0.255) |
| Dec.-Trans. | 0.255 | 0.042 | 535.584 | 0.0 | (0.113, 0.407) |
| Switch-Trans. | 0.292 | 0.037 | 402.528 | 0.0 | (0.151, 0.451) |

Table 7: Tests for the Markov property of the learned probability distributions. The average Jensen-Shannon Divergence across samples is presented for trained models and non-trained models. Closer to 0 means more similar. T-tests were conducted between the two models to check whether model is learning, along with 0.95 confidence intervals on the means. $N = 6388$ samples drawn from the training and test distribution (80/20 split) for these tests.

| Dataset | Trained Model | Untrained Model |
|---|---|---|
| Permutation Data | (1.078, 1.155) | (0.498, 0.534) |
| Random Data | (1.453, 1.557) | (0.484, 0.519) |

Table 8: 95% Confidence Intervals on Variance using Chi-Square Test for NADE Models

| Dataset | Confidence Interval | Credibility Interval | HDI Interval |
|---|---|---|---|
| Trained Permutation | (-0.051, 0.058) | (-0.050, 0.058) | (-0.050, 0.058) |
| Trained Random | (-0.083, 0.064) | (-0.083, 0.064) | (-0.083, 0.063) |
| Untrained Permutation | (-0.071, -0.020) | (-0.071, -0.021) | (-0.071, -0.021) |
| Untrained Random | (-0.034, 0.015) | (-0.034, 0.015) | (-0.035, 0.014) |

Table 9: Comparison of Different Interval Estimates for NADE Models

| Bounds | Dataset | T-Stat Lower | P-Value Lower | T-Stat Higher | P-Value Higher |
|---|---|---|---|---|---|
| (-0.1, 0.1) | Permutation | 3.706 | 1.09e-4 | -3.465 | 2.73e-4 |
| | Random | 2.409 | 8.05e-3 | -2.910 | 1.83e-3 |
| (-0.25, 0.25) | Permutation | 9.084 | 1.51e-19 | -8.842 | 1.21e-18 |
| | Random | 6.399 | 1.02e-10 | -6.900 | 3.73e-12 |
| (-0.5, 0.5) | Permutation | 18.047 | 1.05e-66 | -17.805 | 3.91e-65 |
| | Random | 13.049 | 2.36e-37 | -13.550 | 5.67e-40 |
| (-1.0, 1.0) | Permutation | 35.973 | 2.30e-208 | -35.731 | 2.79e-206 |
| | Random | 26.348 | 1.22e-127 | -26.849 | 1.17e-131 |

Table 10: Hypothesis Test Results for Trained NADE Model

| Bounds | Dataset | T-Stat Lower | P-Value Lower | T-Stat Higher | P-Value Higher |
|---|---|---|---|---|---|
| (-0.1, 0.1) | Permutation | 4.238 | 1.19e-5 | -11.270 | 1.05e-28 |
| | Random | 7.192 | 4.89e-13 | -8.758 | 2.48e-18 |
| (-0.25, 0.25) | Permutation | 15.869 | 4.48e-53 | -22.902 | 6.60e-101 |
| | Random | 19.155 | 4.43e-74 | -20.720 | 5.82e-85 |
| (-0.5, 0.5) | Permutation | 35.255 | 3.59e-202 | -42.287 | 4.03e-263 |
| | Random | 39.092 | 2.16e-235 | -40.658 | 5.49e-249 |
| (-1.0, 1.0) | Permutation | 74.026 | 0 | -81.058 | 0 |
| | Random | 78.967 | 0 | -80.533 | 0 |

Table 11: Hypothesis Test Results for Untrained NADE Model

| Dataset | Trained Model | Untrained Model |
|---|---|---|
| Permutation Data | (0.665, 0.713) | (1.625, 1.741) |
| Random Data | (6.369, 6.827) | (5.547, 5.946) |

Table 12: 95% Confidence Intervals for Variance Using Chi-Square Test for Encoder-Decoder Transformer

| Dataset | Confidence Interval | Credibility Interval | HDI Interval |
|---|---|---|---|
| Trained Permutation | (-0.090, -0.038) | (-0.090, -0.038) | (-0.091, -0.039) |
| Trained Random | (-0.247, 0.399) | (-0.239, 0.394) | (-0.244, 0.387) |
| Untrained Permutation | (-0.358, -0.126) | (-0.359, -0.125) | (-0.362, -0.129) |
| Untrained Random | (-0.419, 0.263) | (-0.413, 0.260) | (-0.416, 0.257) |

Table 13: Comparison of Different Interval Types Across Models for Encoder-Decoder Transformer

| Bounds | Dataset | T-Stat Lower | P-Value Lower | T-Stat Higher | P-Value Higher |
|---|---|---|---|---|---|
| (-0.1, 0.1) | Permutation | 2.712 | 3.38e-3 | -12.408 | 4.04e-34 |
| | Random | 1.069 | 0.143 | -0.144 | 0.443 |
| (-0.25, 0.25) | Permutation | 14.053 | 1.11e-42 | -23.749 | 2.54e-107 |
| | Random | 1.979 | 0.024 | -1.054 | 0.146 |
| (-0.5, 0.5) | Permutation | 32.954 | 1.93e-182 | -42.650 | 2.88e-266 |
| | Random | 3.495 | 2.44e-4 | -2.571 | 5.12e-3 |
| (-1.0, 1.0) | Permutation | 70.756 | 0 | -80.452 | 0 |
| | Random | 6.528 | 4.47e-11 | -5.604 | 1.23e-8 |

Table 14: Hypothesis Test Results for Trained Model Encoder-Decoder Transformer

| Bounds | Dataset | T-Stat Lower | P-Value Lower | T-Stat Higher | P-Value Higher |
|---|---|---|---|---|---|
| (-0.1, 0.1) | Permutation | 27.115 | 8.36e-134 | 22.359 | 1.000 |
| | Random | 0.523 | 0.300 | -0.870 | 0.192 |
| (-0.25, 0.25) | Permutation | 30.682 | 3.25e-163 | 18.792 | 1.000 |
| | Random | 1.568 | 0.059 | -1.915 | 0.028 |
| (-0.5, 0.5) | Permutation | 36.627 | 5.03e-214 | 12.847 | 1.000 |
| | Random | 3.309 | 4.79e-4 | -3.656 | 1.32e-4 |
| (-1.0, 1.0) | Permutation | 48.518 | 9.59e-317 | 0.956 | 0.830 |
| | Random | 6.791 | 7.82e-12 | -7.138 | 7.16e-13 |

Table 15: Hypothesis Test Results for Untrained Model Encoder-Decoder Transformer

| Dataset | Trained Model | Untrained Model |
|---|---|---|
| Permutation Data | (0.511, 0.548) | (2.285, 2.449) |
| Random Data | (6.639, 7.116) | (6.717, 7.200) |

Table 16: 95% Confidence Intervals for Variance Using Chi-Square Test for Decoder-only Transformer

| Dataset | Confidence Interval | Credibility Interval | HDI Interval |
|---|---|---|---|
| Trained Permutation | (-0.158, -0.090) | (-0.157, -0.090) | (-0.158, -0.090) |
| Trained Random | (-0.261, 0.413) | (-0.259, 0.411) | (-0.251, 0.418) |
| Untrained Permutation | (-0.358, -0.126) | (-0.360, -0.125) | (-0.359, -0.124) |
| Untrained Random | (-0.419, 0.263) | (-0.415, 0.256) | (-0.413, 0.257) |

Table 17: Comparison of Different Interval Types Across Models for Decoder-only Transformer

| Bounds | Dataset | T-Stat Lower | P-Value Lower | T-Stat Higher | P-Value Higher |
|---|---|---|---|---|---|
| (-0.1, 0.1) | Permutation | -1.386 | 0.917 | -13.007 | 3.85e-37 |
| | Random | 1.024 | 0.153 | -0.140 | 0.445 |
| (-0.25, 0.25) | Permutation | 7.330 | 1.82e-13 | -21.724 | 3.45e-92 |
| | Random | 1.897 | 0.029 | -1.012 | 0.156 |
| (-0.5, 0.5) | Permutation | 21.857 | 3.64e-93 | -36.250 | 9.16e-211 |
| | Random | 3.352 | 4.11e-4 | -2.467 | 6.86e-3 |
| (-1.0, 1.0) | Permutation | 50.911 | 0 | -65.304 | 0 |
| | Random | 6.261 | 2.45e-10 | -5.377 | 4.35e-8 |

Table 18: Hypothesis Test Results for Trained Model Decoder-only Transformer

| Bounds | Dataset | T-Stat Lower | P-Value Lower | T-Stat Higher | P-Value Higher |
|---|---|---|---|---|---|
| (-0.1, 0.1) | Permutation | -2.405 | 0.992 | -5.787 | 4.30e-9 |
| | Random | 0.126 | 0.450 | -1.024 | 0.153 |
| (-0.25, 0.25) | Permutation | 0.132 | 0.447 | -8.324 | 9.01e-17 |
| | Random | 0.989 | 0.161 | -1.887 | 0.030 |
| (-0.5, 0.5) | Permutation | 4.360 | 6.91e-6 | -12.552 | 7.78e-35 |
| | Random | 2.427 | 7.66e-3 | -3.325 | 4.53e-4 |
| (-1.0, 1.0) | Permutation | 12.817 | 3.63e-36 | -21.008 | 5.13e-87 |
| | Random | 5.303 | 6.49e-8 | -6.200 | 3.58e-10 |

Table 19: Hypothesis Test Results for Untrained Model Decoder-only Transformer

| Dataset | Trained Model | Untrained Model |
|---|---|---|
| Permutation Data | (0.234, 0.250) | (2.831, 3.034) |
| Random Data | (6.414, 6.875) | (8.640, 9.261) |

Table 20: 95% Confidence Intervals for Variance Using Chi-Square Test for Switch Transformer

| Dataset | Confidence Interval | Credibility Interval | HDI Interval |
|---|---|---|---|
| Trained Permutation | (0.002, 0.026) | (0.003, 0.026) | (0.003, 0.026) |
| Trained Random | (-0.226, 0.424) | (-0.222, 0.423) | (-0.235, 0.405) |
| Untrained Permutation | (-0.210, 0.077) | (-0.209, 0.078) | (-0.210, 0.078) |
| Untrained Random | (-0.369, 0.507) | (-0.360, 0.497) | (-0.367, 0.488) |

Table 21: Comparison of Different Interval Types Across Models for Switch Transformer

| Bounds | Dataset | T-Stat Lower | P-Value Lower | T-Stat Higher | P-Value Higher |
|---|---|---|---|---|---|
| (-0.1, 0.1) | Permutation | 18.895 | 2.50e-72 | -14.179 | 2.26e-43 |
| | Random | 1.198 | 0.116 | -0.007 | 0.497 |
| (-0.25, 0.25) | Permutation | 43.701 | 2.31e-275 | -38.985 | 1.85e-234 |
| | Random | 2.101 | 0.018 | -0.910 | 0.181 |
| (-0.5, 0.5) | Permutation | 85.043 | 0 | -80.327 | 0 |
| | Random | 3.607 | 1.60e-4 | -2.416 | 7.90e-3 |
| (-1.0, 1.0) | Permutation | 167.729 | 0 | -163.013 | 0 |
| | Random | 6.619 | 2.46e-11 | -5.428 | 3.29e-8 |

Table 22: Hypothesis Test Results for Trained Model Switch Transformer

| Bounds | Dataset | T-Stat Lower | P-Value Lower | T-Stat Higher | P-Value Higher |
|---|---|---|---|---|---|
| (-0.1, 0.1) | Permutation | 0.459 | 0.323 | -2.270 | 0.012 |
| | Random | 0.755 | 0.225 | -0.139 | 0.445 |
| (-0.25, 0.25) | Permutation | 2.506 | 6.15e-3 | -4.317 | 8.38e-6 |
| | Random | 1.426 | 0.077 | -0.810 | 0.209 |
| (-0.5, 0.5) | Permutation | 5.918 | 1.99e-9 | -7.729 | 9.50e-15 |
| | Random | 2.544 | 5.53e-3 | -1.928 | 0.027 |
| (-1.0, 1.0) | Permutation | 12.742 | 8.64e-36 | -14.553 | 1.89e-45 |
| | Random | 4.779 | 9.60e-7 | -4.164 | 1.65e-5 |

Table 23: Hypothesis Test Results for Untrained Model Switch Transformer

| Model | Train $\mu$ | Train $\sigma$ | 0.95 CI Train | No-train $\mu$ | No-train $\sigma$ | 0.95 CI No-train |
|---|---|---|---|---|---|---|
| Transformer | 13.74 | 1.52 | (13.67, 13.82) | -72.80 | 6.37 | (-73.11, -72.49) |
| Decoder-Trans. | 13.54 | 1.42 | (13.47, 13.61) | -49.70 | 8.58 | (-50.12, -49.28) |
| Switch-Trans. | 13.96 | 1.57 | (13.88, 14.04) | -74.74 | 7.23 | (-75.10, -74.39) |

Table 24: Tests for comparing the difference in the log probability assigned by a model and the true log probability of the sequence (LD). We observe that all three model architectures show significant $> 1$ average differences with large standard deviations. We also observe that this differs markedly from the non-trained models, which are significantly underconfident relative to the true probabilities.

# E   Higher-Order Markov Chain Results

| Dataset | Trained Model | Untrained Model |
|---|---|---|
| Permutation Data | (0.701, 0.751) | (1.123, 1.204) |
| Random Data | (8.352, 8.952) | (6.412, 6.873) |

Table 25: 95% Confidence Intervals for Variance Using Chi-Square Test for Higher-Order Transformer

| Dataset | Confidence Interval | Credibility Interval | HDI Interval |
|---|---|---|---|
| Trained Permutation | (-0.051, 0.020) | (-0.050, 0.020) | (-0.050, 0.020) |
| Trained Random | (-0.440, 0.407) | (-0.435, 0.399) | (-0.428, 0.403) |
| Untrained Permutation | (0.125, 0.239) | (0.124, 0.238) | (0.125, 0.239) |
| Untrained Random | (-0.345, 0.306) | (-0.348, 0.305) | (-0.342, 0.309) |

Table 26: Comparison of Different Interval Types Across Models for Higher-Order Transformer

| Bounds | Dataset | T-Stat Lower | P-Value Lower | T-Stat Higher | P-Value Higher |
|---|---|---|---|---|---|
| (-0.1, 0.1) | Permutation | 4.666 | 1.66e-6 | -6.362 | 1.30e-10 |
| | Random | 0.386 | 0.350 | -0.540 | 0.295 |
| (-0.25, 0.25) | Permutation | 12.937 | 8.84e-37 | -14.633 | 6.73e-46 |
| | Random | 1.080 | 0.140 | -1.234 | 0.109 |
| (-0.5, 0.5) | Permutation | 26.722 | 1.23e-130 | -28.418 | 2.07e-144 |
| | Random | 2.236 | 0.013 | -2.390 | 8.48e-3 |
| (-1.0, 1.0) | Permutation | 54.292 | 0 | -55.988 | 0 |
| | Random | 4.549 | 2.90e-6 | -4.703 | 1.39e-6 |

Table 27: Hypothesis Test Results for Trained Model Higher-Order Transformer

| Bounds | Dataset | T-stat Lower | P-value Lower | T-stat Higher | P-value Higher |
|---|---|---|---|---|---|
| (-0.1, 0.1) | Permutation | 9.692 | 6.33e-22 | 2.813 | 0.998 |
| | Random | 0.486 | 0.314 | -0.719 | 0.236 |
| (-0.25, 0.25) | Permutation | 14.851 | 3.92e-47 | -2.346 | 9.56e-3 |
| | Random | 1.390 | 0.082 | -1.623 | 0.052 |
| (-0.5, 0.5) | Permutation | 23.450 | 4.83e-105 | -10.944 | 3.12e-27 |
| | Random | 2.896 | 1.91e-3 | -3.129 | 8.92e-4 |
| (-1.0, 1.0) | Permutation | 40.647 | 6.89e-249 | -28.141 | 3.84e-142 |
| | Random | 5.909 | 2.10e-9 | -6.142 | 5.13e-10 |

Table 28: Hypothesis Test Results for Untrained Model Higher-Order Transformer

