# OpenReview forum: "Models with a Cause: Causal Discovery with Language Models on Temporally Ordered Text Data"
_TMLR — Accepted by TMLR_

### Review · Reviewer_wBs4 · 2025-12-22

**Summary Of Contributions:**

**Summary** - The paper explores whether Language Models possess the inductive biases required to identify the causal structures governing token generation processes. By focusing on token-level causation rather than semantic fact extraction, the authors provide a mathematical proof that the temporal ordering of human language allows for the unique recovery of causal models, serving as a critical tie-breaker for determining the direction of influence between variables. Through experiments with synthetic Markov chain mixtures, the authors demonstrate that while LMs fail to approximate exact mathematical probability distributions, they successfully learn the necessary statistical properties.

**Strengths**
1. The work provides a strong formal proof (proposition 4.2) to show that temporal ordering enables recovery of causal structures under standard assumptions
2. It addresses a major limitation of traditional causal discovery algorithms which scale poorly to high-dimensional nature of text.
3. Presents two specific biases required for discovery - learning conditional independencies and ability to detect temporal order
4. The work provide a realistic look at how LMs learn. Discovering they rely on ordinal rankings rather than exact distribution approximation is a significant insight into the internal logic of these models
5. Multiple architectural variants are tested

**Weaknesses**
1. Proposition 4.2 requires that each unobserved variable $U$ has at most one child. In natural text, unobserved "confounders" like the topic of a conversation or the author's mood act as parents to many tokens simultaneously. By restricting unobserved variables to only one child, it seems like the work is testing a "best-case" scenario that doesn't account for the complex hidden influences typical of human speech.
2. The paper relies on the Faithfulness condition, which assumes that if we observe statistical independence between two tokens, it's because there is no causal arrow between them.  LMs often learn redundant representations where different internal paths might often cancel each other out, statistically, even if a causal relationship exists, which in my opinion, makes this assumption unlikely to hold in high-dimensional text. And, if the "Faithfulness" assumption is violated, which is common in a complex networks,  the algorithms the authors discuss (PC/IC) can fail to identify the correct structure
3. A major concern is the reliance on Mixtures of Markov Chains as a proxy for human language. Human language is governed by complex hierarchies, long-range dependencies, and deep semantic structures that first-order Markov processes cannot capture.
4. JSD results (the conditional independence test) for untrained models seems to create a false positive concern, given they show near-zero JSD, which usually indicates perfect conditional independence. Although this might be because of the random weights, it is difficult to determine if a low JSD in a trained model represents a sophisticated understanding of causal structure or just a "residual" effect of this baseline architectural tendency. Is this a good metric for this problem?
5. Causal discovery properties are not universal. Decoder-only models only partially learned Markov exchangeability compared to Encoder-Decoder models. this might indicate that the causal bias is not a universal feature of all LMs, but is highly sensitive to the specific architecture.
6. Minor typos - (a) in definition 3.1, the functions in the set should be indexed as $f_i$ for each ith variable; (b) the random vector init. $X_i$ interchangeably uses $\omega$ and w

**Audience:**

Yes

**Audience Explanation:**

The paper tackles foundational questions about inductive bias, identifiability, and causal discovery in language models. Even with idealized assumptions, the combination of formal guarantees, controlled experiments, and the findings can offer insights that are valuable to researchers studying causality, sequential modeling, and the internal behavior of LMs, even if the results do not directly transfer to natural language settings.

**Broader Impact Concerns:**

No major concerns on the ethical or societal implications of the work.

**Claims And Evidence:**

Yes

**Claims Explanation:**

The theoretical claim that temporal ordering enables unique causal recovery and the empirical claim that some transformer architectures learn probabilistic symmetries like Markov exchangeability on synthetic data, are well supported. However, several results rely on strong assumptions (reliance on mixture of Markov chains, faithfulness, limited latent confounders) that are not indicative of natural language, and some of the metrics (conditional independence tests using JSD), are ambiguous, which makes the claims a little less convincing.

**Requested Changes:**

1. Update and standardizing the notations - this will help in readability (strengthening)
2. For the single child assumption, please add a discussion section addressing how the theory holds up when unobserved variables have multiple children, or conduct an experiment with shared latent variables (confounder) to test model robustness. (critical)
3. Regarding the faithfulness assumption, it would be helpful if it is reframed as a research question - "To what extent do high-dimensional LMs satisfy faithfulness?" or include a sensitivity analysis on faithfulness violations. (critical)
4. Please add an experiment using a more complex synthetic grammar (like a context free grammar) to see if the causal discovery biases persist beyond simple Markovian transitions. (critical)
5. Clarify that JSD alone is an insufficient metric (or provide reasons to support) and provide a more robust baseline that distinguishes ignoring context from recognizing conditional independence (critical)
6. Expand the discussion on how "ordinal" knowledge limits the model's ability to perform "cardinal" tasks like causal effect estimation (measuring the magnitude of a cause) (critical)
7. Please add a simple experiment or an analysis on whether calibrating the models (making their prob more accurate) helps or hurts their ability to discover causal structures (strengthening)
8. Add a discussion on why Decoder-only architectures , the most popular type for current LLMs, struggle to learn markov exchangeability more than the other variants (strengthening)

---

> ### Author Response · Authors · 2026-01-05
> **Response to Reviewer**
>
> We thank the reviewer for their thorough and constructive feedback. We address each point below.
>
> ## Weaknesses
>
> **W1 (Single-child assumption):** We agree this represents a best-case scenario. We adopt this deliberately: if models cannot perform discovery under favorable conditions, they cannot succeed in more complex settings. We have expanded the limitations section to discuss causal sufficiency and future work on algorithms like FCI for non-unique discovery.
>
> **W2 (Faithfulness):** We appreciate this concern but note it conflates two distinct problems. Our work tests whether LMs can discover structure from a *generating* distribution that satisfies faithfulness—not whether the LM's internal representations are faithful. The worry about redundant representations canceling out applies to extracting an SCM from the model, which we leave to future work. We have clarified this distinction in the limitations section.
>
> **W3 (Markov chains as proxy):** We address this with new experiments on 2nd-order Markov processes (see below). Regarding context-free grammars: CFG rules act as latent confounders across tokens, violating causal sufficiency. Higher-order Markov chains test longer dependencies while preserving the theoretical conditions of Proposition 4.2.
>
> **W4 (JSD false positives):** The reviewer correctly identifies that untrained models show near-zero JSD. This motivates our Markov exchangeability tests, which do discriminate trained from untrained models. We have added discussion of JSD's limitations and note that trained models show measurable drift from untrained baselines.
>
> **W5 (Architecture differences):** Multi-seed experiments (5 seeds per architecture) now show tighter results. Decoder-only transformers improved substantially, performing only slightly below encoder-decoder models.
>
> **W6:** Fixed. Thank you.
>
> ## Requested Changes
>
> **RC1 (Notation):** Standardized throughout.
>
> **RC2-3 (Sufficiency/Faithfulness):** Added expanded limitations discussion addressing both concerns.
>
> **RC4 (Complex grammar):** We conducted 2nd-order Markov experiments on encoder-decoder transformers:
>
> | Model | Var. CI (Perm.) | CI (Perm.) | CI (Random) |
> |-------|-----------------|------------|-------------|
> | Trained | (0.701, 0.751) | (-0.051, 0.020) | (-0.440, 0.407) |
> | Untrained | (1.123, 1.204) | (0.125, 0.239) | (-0.345, 0.306) |
>
> Trained models show tight confidence intervals around zero with variance below 1, validating learning of longer-range dependencies (results reported in appendix E).
>
> **RC5 (JSD metric):** Added discussion of underdetermination. Multi-seed results (5 models each) (results changed in table 2):
>
> | Model | JSD-Train 0.95 CI | JSD-NT μ |
> |-------|-------------------|----------|
> | NADE | (0.012, 0.013) | 0.002 |
> | Enc.-Dec.-T. | (0.143, 0.147) | 0.018 |
> | Decoder-T. | (0.297, 0.301) | 0.057 |
> | Switch-T. | (0.261, 0.265) | 0.051 |
>
> **RC6 (Ordinal/Cardinal):** Added limitations discussion on how ordinal rankings are insufficient for causal effect estimation requiring magnitude comparisons.
>
> **RC7 (Calibration):** Added discussion arguing better-calibrated probabilities would improve effect estimation in limitations section.
>
> **RC8 (Decoder-only):** Multi-seed experiments show improved decoder-only performance (results updated in figure 2 and appendix D):
>
> | Dataset | Confidence Interval |
> |---------|---------------------|
> | Var. - Trained Perm. | (0.511, 0.548) |
> | Var. - Untrained Perm. | (2.285, 2.449) |
> | Trained Perm. | (-0.158, -0.090) |
> | Trained Random | (-0.261, 0.413) |
>
> These results indicate decoder-only architectures do learn Markov exchangeability, though with slightly higher variance than encoder-decoder models.
>
> We believe these revisions substantially strengthen the paper and thank the reviewer for their valuable suggestions, and we have updated the manuscript accordingly.

---

> > ### Author Response · Authors · 2026-02-11
> > **Updated Manuscript and Comments**
> >
> > We have updated the manuscript as described by the previous comments, which should now be visible.

---

> ### Author Response · Authors · 2026-02-19
> **Follow Up To Response**
>
> We thank the reviewer for their thoughtful review. We wanted to follow up to ensure we have adequately addressed the reviewers concerns. The initial comments we posted in reply to their review were given before the rebuttal period began in [this comment](https://openreview.net/forum?id=YJddclPGuY&noteId=gGCH3hTJgs), which was not initially visible. After the response phase began, we made it visible and posted an additional comment but this may not have triggered a notification.
>
> To summarize our comments, we have revised the manuscript to address the recommendations and criticisms the reviewer identified:
>
> 1. Notation was standardized throughout.
> 2. Added expanded discussion of sufficiency and faithfulness.
> 3. We conducted second-order Markov experiments to validate identification of longer dependencies.
> 4. Across all experiments we ran multi-seed experiments to better validate JSD and Decoder-only transformer results.
> 5. We added extended discussions in the limitations section and elsewhere to better address the specific weaknesses and worries of the reviewer.
>
> We would greatly appreciate the reviewer's feedback on our response along with the updated draft at their earliest convenience.
>
> Thank you for your time.

---

> > ### Comment · Reviewer_wBs4 · 2026-02-20
> >
> > I would like to thank the authors for their response and clarifying the concerns, and for carrying out the additional experiments. The inclusion of the 2nd-order Markov tests and the expanded limitations discussion strengthens the paper's grounding. I have one suggestion. In the main text where specific assumptions (like causal sufficiency or faithfulness) are defined, please include a brief forward-reference to the Limitations section. This would ensure that the readers are immediately aware of the boundary conditions of the findings.
> >
> > Some additional pointers. It would be extremely useful if the updates made to the manuscript are highlighted (pdf with tracked changes or some highlighting). Otherwise it is challenging to review the changes that have been made. Additionally, in your responses, please explicitly map the new results/tables to their specific section/table numbers in the updated manuscript

---

> > > ### Author Response · Authors · 2026-02-20
> > > **Response to Further Comments**
> > >
> > > Thank you for kind remarks and feedback. We believe this has further improved the paper.
> > >
> > > We added the following parenthetical remark at the end of the preliminaries section after the definition of faithfulness and sufficiency:
> > >
> > > > (we discuss how these assumptions put limitations on our methods in section 7)
> > >
> > > Additionally, we have added an annotated manuscript that includes all changes between the first draft you read and the current draft. You will find that manuscript as the current revision. The changes highlighted there include the revisions made to all tables, though the figures lack those highlights (they are the new figures per requested by the other reviewers). We have updated the tables in our previous comment to reflect which tables and figures use the new values gathered by our multi-seed experiments. As mentioned, you should for the tables see all the exact changes in our annotated manuscript.
> > >
> > > Thank you for your feedback.

---

### Review · Reviewer_4kqN · 2025-12-24

**Summary Of Contributions:**

The authors investigate the potential induced biases in LLMs that affect their ability to detect temporally ordered causal structures in texts. They show that the true, fully directed, causal graph can be recovered from data and a given total ordering of the variables under the assumptions of Markovianity, faithfulness, and sufficiency. The authors conclude that, for LLMs, to perform this task, they need to possess the ability to detect independencies and temporal orderings in texts and argue that evaluating independencies alone might not be sufficient and therefore resort to measuring Markov exchangeability.

In the experimental section, the authors consider mixtures of Markov chains to conduct initial experiments on synthetic data. They find that transformers partially adapt to the underlying Markov process, while not fully adhering to its conditional independence structure. The authors furthermore test whether model predictions yield Markov exchangeable properties. Results suggest that encoder-decoder and switch transformers appear to adapt to Markov exchangeability. While absolute probabilities seem to be systematically overestimated, the authors find that qualitative rankings show some correlation with random baselines.

[Disclaimer: I reviewed this paper before. Since the presented version appears to align with its prior version to a large degree, I largely retain my initial review. I still acknowledge the changed scope and focus of this work as a submission to TMLR and have adjusted the relevant parts accordingly.]

**Additional Comments:**

I find the paper to be generally well-written and well-executed. The authors present an interesting new perspective on Markov exchangability and connect it to the learning/inference process on mixtures of Markov processes. While the authors focus on the very particular setting of a (causal/Markov) chain, the obtained results yield insights into the distributions and mechanisms learned by the model.

**Exchangeability and Causal Structure.** My main concern still regards the connection between learning conditional independencies and Markov exchangeability. While I can see the broader connection between exchangeability and the (possibly causal/structural) transition dynamics, I fail to see it on a mathematical level for the particular questions asked in the paper.

1. Could the authors further detail the relation between inferring the presence of causal mechanisms as discussed in this paper and the applied exchangeability measure (e.g., also concerning my comments in the 'Claims' section)?
2. Similarly, I would find it helpful if the authors could detail more precisely the process of transforming random samples with Markov-exchangeable permutations.

**Audience:**

Yes

**Audience Explanation:**

Multiple existing works have been concerned with the effects of non-temporal orderings deterring LLMs' causal discovery abilities. To the best of my knowledge, and although it is a common belief in the field, the consistency of reliable causal discovery in the contra-positive case has not been explicitly examined so far. The presented paper may therefore serve as a good reference to support this common belief and elucidate the capabilities and limitations of LLMs on this task.

Theory and experiments aim to empirically quantify the effects of data on learned mechanisms by measuring the resulting output probabilities. Experiments are restricted to a very specific setting, where learning mixed causal Markov transitions is performed from a synthetically generated symbolic dataset. Particularly, the authors could have placed a stronger emphasis on why the evaluated metrics target causal rather than correlational sequences.

**Claims And Evidence:**

Yes

**Claims Explanation:**

The presentation and handling of theorems and proofs in the paper seem to be sound. Data creation, experimental setup, and evaluations are well described and carried out. The interpretation of the results mostly aligns with the obtained numbers.

**Exchangeability.** While the authors test for Markov exchangeability in their setup, I am not particularly sure, how the process tested in Section 5 can be made Markov exchangeable, as he underliying data generation process seems to be still dependent on the  Markov process, such that Eq. 5 holds, where $X_{i-1}$ is generally not exchangeable when predicting $X_i$. Although the authors claim to generate Markov-exchangeable permutations from a set of samples, no details are provided, nor are any examples shown. While this may be a lack of understanding on my part, I would like to ask the authors to further elaborate on this connection in their paper.

**Data Dependence of Untrained Models.** In Section 5.4, the authors state, "Untrained models show the opposite pattern of slightly decreasing confidence as dataset size increases". I fail to see how untrained/randomly initialized models can make any meaningful predictions and, therefore, serve as comparative baselines. The authors should exclude this observation to be an empirical artifact.

**Requested Changes:**

Proposition 4.2 --that total temporal orderings are sufficient to direct partially directed graphs (or any correlational skeleton, for that matter)-- is a known result (e.g., [1]; Incorporating Background Knowledge or [2]; Sec. 7.5.1). Even though the authors draw further insights on the necessity of LLMs to detect independency and temporal ordering statements from it, the authors should mention this fact.

**Minor Comments and Typos:**
- The plots in Figure 2 are hard to read. The authors might want to reduce margins and increase the font size.
- The authors might want to reference their added proof of Proposition 4.2 in the appendix from within the main text.


[1] P., Glymour, C., & Scheines, R. (2000). Causation, Prediction, and Search. MIT press.
[2] Pearl, J., 2009. *Causality*. Cambridge university press.

---

> ### Author Response · Authors · 2026-01-05
> **Response to Reviewer's Comments**
>
> We thank the reviewer for their careful and constructive evaluation of our manuscript. Our replies are below.
>
> ## Major Comments
>
> ### Exchangeability and Data Generation
>
> > *"I am not particularly sure how the process tested in Section 5 can be made Markov exchangeable ... "*
>
> Markov exchangeability concerns transition counts: two sequences are n-Markov exchangeable if they share identical transition counts between n-blocks. A probability function assigning equal probabilities to such sequences is representable as a mixture of Markov processes.
>
> Our data generation samples from a prior $\Pr(X_1)$, which determines a stochastic matrix; we then sample from this matrix to generate remaining tokens. This is by definition a mixture of Markov chains. Our test evaluates whether models assign equal probabilities to sequences sharing the same starting value and transition counts—precisely the condition for learning the mixture structure.
>
> We have added clarifying text to Section 5:
>
> To clarify this, we have added the following to the start of Section 5:
>
> > "We generate each sequence by first sampling from a prior distribution $\Pr(X_{1})$ ... This makes the generating process a mixture of Markov chains as given in figure 1."
>
> And this to the start of the symmetry tests:
>
> > "Assigning equal probabilities between sequences with the same starting sequence value and transition counts implies the model probability function has identified the generating mixture Markov chain. Successful learning of Markov exchangeability across all chains implies learning of the full generating mixture model."
>
> Regarding permutation generation: we omitted details for space, but the basic method is a backtracking search constrained by the transition count matrix. We fix the starting element, then search over permutations that preserve the original sequence's transition counts. We have added the following clarification:
>
> > "Markov-exchangeable permutations are generated from the transition count matrix of a sequence: we perform a backtrack search over sequences that conform to the transition counts of the original sequence we aim to permute."
>
> ### Data Dependence of Untrained Models
>
> > *"I fail to see how untrained/randomly initialized models can make any meaningful predictions and, therefore, serve as comparative baselines. The authors should exclude this observation to be an empirical artifact."*
>
> We agree this observation warrants caution. We have revised the relevant passage to:
>
> > "Untrained models show the opposite pattern of slightly decreasing confidence as dataset size increases, though we believe the decrease is an empirical artifact."
>
> ## Attribution of Proposition 4.2
>
> > *"Proposition 4.2—that total temporal orderings are sufficient to direct partially directed graphs—is a known result (e.g., [1]; Incorporating Background Knowledge or [2]; Sec. 7.5.1). Even though the authors draw further insights on the necessity of LLMs to detect independency and temporal ordering statements from it, the authors should mention this fact."*
>
> We thank the reviewer for this important point. We have added the following acknowledgment immediately after the proposition:
>
> > "Thus, appropriate assumptions enable unique recovery of causal directions (as was suggested by Spirtes et al. (2001, p.93) and is well known)."
>
> We have also added a sentence referencing the appendix proof:
>
> > "The proof is in appendix A."
>
> ## Minor Comments and Typos
>
> > *"The plots in Figure 2 are hard to read. The authors might want to reduce margins and increase the font size."*
>
> We have revised the figures by reducing margins and making additional changes for readability. Note that the figures also reflect updated experiments (now 5-run averages across different random seeds) in response to another reviewer's suggestion for more robust results.
>
> ## Additional Comments: Exchangeability and Causal Structure
>
> > *"My main concern still regards the connection between learning conditional independencies and Markov exchangeability..."*
>
> The connection is as follows. Markov-exchangeable probability functions are representable as mixtures of Markov processes (Diaconis & Freedman, 1980). Our data is generated by exactly such a mixture. The Markov exchangeability test thus evaluates whether models have learned the structure of this generating process—without requiring them to identify the exact mixture components.
>
> This test complements the conditional independence tests. While conditional independence tests check whether models learn the factorization structure (each token depends only on its predecessor and initial token), Markov exchangeability tests check whether models learn the symmetry properties that characterize the mixture-of-Markov-chains family. Together, they provide converging evidence that models can perform limited causal discovery at the token level.
>
> We hope these revisions and clarifications address the reviewer's concerns and have updated the manuscript.

---

> > ### Author Response · Authors · 2026-02-11
> > **Updated Manuscript and Comments**
> >
> > We have updated the manuscript as described by the previous comments, which should now be visible.

---

> ### Author Response · Authors · 2026-02-19
> **Follow Up on Response**
>
> Thank you for your thoughtful review. We wanted to follow up to ensure we have adequately addressed your concerns. Our initial comments, found in [this comment](https://openreview.net/forum?id=YJddclPGuY&noteId=qADyTw1PJ7), were posted prior to the start of the rebuttal period and may not have triggered a notification. We have made it visible as well as updated the manuscript to address your critiques.
>
> To summarize, we have added clarifications into the manuscript about the data generation process, updated our discussion of the importance concerning untrained model results, referenced how our proposition is know, and addressed the connection between conditional independence and exchangeability tests. We would greatly appreciate your feedback on our response and updated draft at your convenience.
>
> Thank you for your time.

---

> > ### Comment · Reviewer_4kqN · 2026-02-19
> >
> > I would like to thank the authors for their detailed reply.
> >
> > The previously omitted detail of a backtracking search for filtering for Markov-exchangeable conformant sequences was *the* important remark that helped me understand the methodology of the paper. Previously, I was assuming that the authors were trying to establish a connection between Markov exchangeability and freely sampled sequences, which would be clearly difficult to achieve. Thank you for clarifying this point.
> >
> > Considering all further answers to my questions, the implemented changes, as well as the additional results on the other reviewer's comments, I am generally positive towards the acceptance of this paper.

---

### Review · Reviewer_G4Zz · 2026-02-10

**Summary Of Contributions:**

This paper investigates if and how LLMs leverage inductive biases to discover the causal structure in autoregressive generation. The authors offer a simple solution for this problem and theoretically demonstrate that the temporal ordering in text can lead to unique "causal model recovery" under their assumptions. That is, LLMs can leverage sequential processing and positional encodings to estimate the relative ranking of what comes next.  Through experiments on synthetic data, they show that transformer variants can learn the conditional independencies and Markov exchangeability properties.

Pros:

1. The paper provides a theoretical proof that LLMs can recover causal structure when token sequences have temporal ordering and satisfy some standard causal assumptions.

2. With synthetic data generated from Markov chains, the authors created controlled experiments where the ground truth was known. This allowed us to evaluate if the models were actually learning the necessary causal structures.

Cons:

1. The paper focuses only on the causal structure of how tokens are generated rather than semantically causal claims in text, such as "smoking causes cancer", meaning the findings may not fully apply to real-world language data. In addition, the theory relies on certain assumptions (Markov etc.) which are unlikely for the actual text corpus.

**Audience:**

Yes

**Audience Explanation:**

Yes, the paper provide some interesting analysis that should be generally of ML/NLP communities' interest.

**Claims And Evidence:**

Yes

**Claims Explanation:**

Yes, the claims are generally supported by convincing evidence regarding the mechanism of learning, but limited to the extend of observing these solely on synthetic data.

**Requested Changes:**

The paper proves LMs can learn causal structures for synthetic data; however, it would be valuable to extend this to real-world cases like math or coding benchmarks, which also exhibit levels of causality.

---

> ### Author Response · Authors · 2026-02-11
> **Response to Reviewer's Comments**
>
> We thank the reviewer for their thoughtful and constructive comments.
>
> We agree that the current work focuses on the causal structure of token generation rather than semantic causal claims in text, and we appreciate the reviewer highlighting this distinction. Our intent was to establish a precise theoretical foundation; we wanted to show under what minimal conditions LMs could potentially discover causal structures before extending to more complex, real-world settings. We believe this controlled setting is a necessary first step, as it allows us to isolate the mechanisms at play without the confounds introduced by natural language.
>
> Regarding the assumptions (e.g., Markov properties), we acknowledge these are idealized relative to real text corpora. However, we note that such assumptions are standard in the causal discovery literature and serve as a tractable starting point for formal analysis. Relaxing these assumptions is an important direction for future work.
>
> We also appreciate the suggestion to evaluate on real-world benchmarks such as math or coding tasks. We agree this would be a valuable extension, and we discuss it as further avenue for future research in the revised manuscript in the expanded limitations section.

---

### Decision · Action_Editor_SvRq · 2026-04-08

**Recommendation:** Accept as is

**Audience:**

Yes

**Audience Explanation:**

The findings of this paper will be of interest to researchers who work at the intersection of causal discovery and language models, which is an active area of research.

**Claims And Evidence:**

Yes

**Claims Explanation:**

This paper examines how tokens depend on their predecessors by assessing whether they exhibit the temporal and statistical properties required for causal discovery. The central finding is that certain classes of language models possess inductive biases that enable the recovery of causal structure from text data. In particular, when token sequences satisfy standard causal assumptions and admit a temporal ordering, a unique causal model can be identified. These claims are supported by both formal guarantees and controlled experiments.

There is broad agreement among the expert reviewers that the paper addresses fundamental questions concerning inductive bias, identifiability, and causal discovery in language models. While the scope is limited to mixtures of relatively simple Markov chain structures, the problem formulation is well motivated, the experimental evaluation is carefully conducted, and the findings are likely to be of interest to researchers who work at the intersection of causal discovery and language models.